# Convergent Validity of Two Sensory Questionnaires in Spain: Sensory Profile-2 and Sensory Processing Measure

**DOI:** 10.3390/children10091516

**Published:** 2023-09-06

**Authors:** Berta Gándara-Gafo, Isabelle Beaudry-Bellefeuille

**Affiliations:** 1Department of Occupational Therapy, Health Integration and Promotion Research Unit (INTEGRA SAÚDE), Faculty of Health Sciences, University of A Coruña, Campus de Oza, 15006 A Coruña, Spain; 2Clínica de Terapia Ocupacional Pediátrica Beaudry-Bellefeuille, Calle Marqués de Santa Cruz 7, 33007 Oviedo, Spain; ibbergo@gmail.com

**Keywords:** validity, sensory reactivity, occupational therapy

## Abstract

(1) Background: Several sensory questionnaires aimed at analyzing sensory reactivity problems in children are available in Spain; however, knowledge about whether these questionnaires can obtain equivalent results is lacking. The purpose of this study was to examine the convergent validity of two sensory questionnaires available for the Spanish population (Spain): Sensory Profile-2 (SP-2) and Sensory Processing Measure (SPM). (2) Methods: This study involved a sample of 116 children between the ages of 5 and 12 years with sensory integration differences and concerns with participation in daily activities. A Pearson’s correlation coefficient was calculated along with the significance for this test. (3) Results: Correlations between SP-2 and SPM subscales ranged from r = 0.127 (p. 174) to r = 0.674 (*p* < 0.001). Correlations between sensory factors analyzing the same sensory systems ranged from r = 0.401 (*p* < 0.001) to r = 0.674 (*p* < 0.001) for body position/body awareness and hearing, respectively. (4) Conclusions: There is adequate convergent validity between the SPM and the SP-2 for the Spanish population in most sensory factors. The results support the use of both sensory questionnaires with the Spanish population.

## 1. Introduction

Sensory integration theory was developed by A. Jean Ayres [1]. Dr. Ayres’s original investigations sought to understand the functioning of sensory systems in an independent and integrated way. The body of knowledge encompassed within this framework, now referred to as Ayres Sensory Integration (ASI), is aimed at understanding the sensory and motor functions which underlie many aspects of development, social participation, and occupational performance [2]. Inadequate sensory integration may result in difficulties in several areas which are key to performance and participation, including sensory perception; ocular, postural, bilateral integration, and praxis functions; and sensory reactivity [2,3,4,5,6].

Sensory perception is described as the organization and interpretation of sensory input, and it is often followed by a motor act such as postural adjustments or refined motor actions [7]. Sensory reactivity reflects continuous physiological adjustments within the nervous system to ensure adaptation to incoming sensory information. Sensory reactivity problems include hyper and hypo-reactivity [8] that impact participation in activities of daily living [9]. For example, Dr. Ayres linked tactile hyper-reactivity to problems with attention, arousal, activity level, and emotion regulation [1]. 

Sensory reactivity is assessed mostly by using sensory questionnaires; the most widely used in Spain are the Sensory Profile-2 (SP-2) [10] and the Sensory Processing Measure (SPM) [11], the latter currently revised as Sensory Processing Measure-2 (SPM-2) [12]. Both questionnaires are designed to be used together with other evaluation tools, personal narratives as well as clinical observations, as part of a comprehensive evaluation that analyzes sensory integration functions and their impact on participation. These questionnaires have been developed in the United States, and translations are available for use in Spanish-speaking populations. Additionally, the SP-2 [10] has normative data for Spain.

The SP-2 [10] and the SPM [11] have a home and school version. The two questionnaires do not measure all the same constructs, nor do they follow the same theoretical model; the SP-2 is built on the sensory processing model of Dunn [13], whereas the SPM is closely aligned with the constructs and pattens identified within ASI [14]. However, both questionnaires seek to obtain information relative to the sensory processing of specific sensory stimuli such as sound, touch, taste, sight, proprioception and vestibular input. 

The SP-2 home version was designed by Dunn [10] to examine sensory processing using 86 items, each related to a sensory system (auditory, visual, movement, touch, body position, oral) or functional skills which are dependent on sensory integrative function (conduct, social emotional and attentional abilities). Items are also classified within four sensory quadrants (registration, seeking, sensitivity, and avoidance), each one related to sensory thresholds (high/low) or types of self-regulation responses (active/passive): (1) registration, children with high neurological thresholds and passive behavioral strategies; (2) seeking, children with a high neurological threshold who actively seek sensory stimuli; (3) sensitivity, children with low neurological thresholds and passive behavior; and (4) avoidance, children with low neurological thresholds that actively limit their exposure to unpleasant sensations.

The SPM [11] home version includes 75 items designed to assess behavioral responses related to vision, hearing, touch, taste & smell, body awareness, and balance & motion, as well as praxis (planning & ideas). The SPM also includes a section on social participation: a functional skill which is dependent on sensory integrative function. The SPM follows the Ayres Sensory integration model proposed by Bundy, Lane and Murray [15], is compatible with the most recent versions of the ASI^®^ model [8] and is designed to identify possible problems of sensory hyper-reactivity (over-response), sensory hypo-reactivity/perception issues (diminished response/poor ability to recognize and interpret sensory stimuli) as well as issues in postural control and praxis.

During the development of the SPM questionnaire [11], a convergent validity analysis was performed with the sensory questionnaires considered the Gold Standard at that time: Short-SP [16] and Infant/Toddler-SP [17]. The original research results show that these instruments have moderate to strong correlations between sections representing the same sensory systems. The convergent validity process of the original study [11] indicated that both questionnaires could be used to test the same sensory constructs in children in the United States of America. However, the authors themselves indicated the need to carry out convergent validity studies of versions of the SPM adapted to other countries. 

Validity is the ability of an assessment tool to measure that construct for which it has been designed. Convergent validity analyzes the degree to which an instrument measures the same construct as another measure considered the Gold Standard (an alternatively equivalent measure that has adequate psychometric data). The methodology [18] indicates that whenever a Gold Standard measure is available, a convergent validity process must be carried out that includes a representative sample as well as the administration, evaluation and comparison of both tools in each individual.

The availability of several assessment tools that analyze the same construct is useful; however, research on convergent validity must also be available to ascertain whether both tools measure the same thing and whether they can be used in a specific population interchangeably. Currently, a convergent validity analysis has not been carried out that analyzes the correlations between the Spanish version of the SPM and SP-2. A strong correlation was expected in those factors that measure the same sensory systems in the SP-2 [10] and in the SPM [11] (auditory and hearing; visual and vision; movement and balance & motion; touch and touch; body position and body awareness; oral and taste & smell, respectively). Accordingly, the aim of this study was to analyze the convergent validity of both questionnaires to determine whether both tools measure the same constructs and if they can be considered comparable for research and clinical use in Spain.

## 2. Materials and Methods

A two-year cross-sectional study was designed to analyze the convergent validity of two sensory questionnaires (SPM and SP-2) available in Spanish. The sample was made up of children between 5 and 12 years old referred to Occupational Therapy for problems participating in their activities of daily living and linked to underlying sensory integration issues. All children were assessed with the SPM and SP-2 as part of a comprehensive Occupational Therapy assessment in a private pediatric Occupational Therapy clinic in Spain. Prior to data collection, participants were asked for permission and informed consent was obtained through the Health Data Protection form required by the Ministry of Health (Spain). Ethical review and approval were waived for this study by the health authorities, because this study was carried out with anonymous information, with the approval of management of the center where the sample was obtained. The study is in accordance with Regulation 2016/679 of the European Parliament and of the Council regarding the protection of personal data of natural persons. This law allows the use of anonymized data with the consent of the families and does not require further approval by an ethics committee. The data obtained were anonymized and were not available to anyone outside the research team. 

Participants: This study involved a convenience sample of 116 children (boys n = 80, 69%) between the ages of 5 and 12 whose parents responded to the SPM and SP-2 as part of a comprehensive Occupational Therapy assessment. The inclusion criteria were the following: aged between 5 and 12 years old; suspected sensory integration problems identified by an occupational therapist trained in Ayres Sensory Integration (ASI); and informed consent given by the parents/guardians through the Health Data Protection document. Children who did not meet the previously mentioned criteria were excluded.

Measures: This study used the Spanish language version of two parent sensory questionnaires: SP-2 [10] and SPM [11]. Aimed at children aged from 3 to 14 years and 11 months, the SP-2 analyzes sensory factors and sensory patterns derived from Dunn’s model [11]: registration, seeking, sensitivity, and avoidance. Among the sensory factors, it analyzes auditory, visual, movement, touch, body position, and oral processing as well as aspects relating to conduct, social–emotional and attentional abilities derived from sensory processing. The SP-2 has adequate psychometric properties and has reference values in the United States and Spain [10]. Internal reliability data, measured by Cronbach’s Alpha, range between 0.60 and 0.93, and test–retest reliability ranges between 0.66 and 0.97 for the four quadrants. The SP-2 has an apparent or logical validity analysis carried out by experts in the field, a criterion validity analysis carried out between this version and previous versions, and a construct validity analysis that differentiates between populations with typical development and populations with dysfunction.

Aimed at children aged 5–12, the SPM [11] analyzes information relating to social participation, vision, hearing, touch, taste & smell, body awareness, balance & motion, and planning & ideas. The SPM provides adequate psychometric data on both reliability and validity, including convergent validity with the SP [19] and Short SP [16], with data ranging from 0.10 to 0.62 [11]. The SPM has reference values in the United States (n = 1.051) in typically developing children.

Procedure: The procedure described by Parham [11] to analyze the convergent validity of the SP-2 [10] and SPM [11] was used in this study with a Spanish population; a correlation analysis between all sensory factors and functions of the SP-2 [10] (auditory, visual, movement, touch, body position, oral, conduct, social–emotional and attentional) with all sensory factors and functions of the SPM [11] (social participation, vision, hearing, touch, taste & smell, body awareness, balance & motion and planning & ideas). A strong correlation was expected in those factors that measure the same sensory systems in the SP-2 [10] and in the SPM [11] (auditory and hearing; visual and vision; movement and balance & motion; touch and touch; body position and body awareness; oral and taste & smell, respectively).

A correlation analysis was conducted using Pearson’s correlation coefficient along with the significance to this test. Pearson’s coefficient gives a variation ranging between −1 and +1, where 0 is the absence of linear relationship between variables while −1 or +1 is a perfect linear relationship, negative or positive [20]. Values between 0 and 0.10 represent an absence of correlation, values between 0.10 and 0.29 represent a weak correlation, values between 0.30 and 0.49 represent a moderate correlation and values above 0.5 represent a strong correlation [21]. Statistical analysis was carried out using the R program (R Development Core Team), version 4.1.3.

## 3. Results

A total of 116 children between 5 and 12 years of age participated in this study. Table 1 shows the characteristics of the sample. 

Pearson’s correlations between subscales ranged from r = 0.127 (p. 174) for movement (SP-2) and taste/smell (SPM) to r = 0.674 (*p* < 0.001) for auditory (SP-2) and hearing (SPM). The Pearson’s correlations obtained from all the sensory factors that analyze the same stimulus ranged from moderate (r = 0.401, *p* < 0.001) for body position (SP-2) and body awareness (SPM) to strong (r = 0.674, *p* < 0.001) for auditory (SP-2) and hearing (SPM).

The results obtained with the Pearson coefficient indicate moderate correlations in the sensory factors that analyze proprioceptive, oral and vestibular information: body position (SP-2) and body awareness (SPM) (r = 0.401, *p* < 0.001); oral (SP-2) and taste & smell (SPM) (r = 427, *p* < 0.001); movement (SP-2) and balance & motion (SPM) (r = 0.450, *p* < 0.001). Regarding the factors that analyze the processing of auditory, visual and tactile information, strong correlations were obtained: visual (SP-2) and vision (SPM) (r = 635, *p* < 0.001); touch (SP-2) and touch (SPM) (r = 666, *p* < 0.001); auditory (SP-2) and hearing (SPM) (r = 0.674, *p* < 0.001). The results obtained for the convergent validity between SP-2 and SPM subscales are presented in Table 2.

## 4. Discussion

This study is the first to provide convergent correlation data between the SP-2 and SPM sensory questionnaires in Spain, both of which analyze sensory integration and processing differences. The results of the present study show that there are moderate (0.30–0.49) to strong (above 0.5) correlations in those sensory factors that analyze the same sensory systems [21]. Specifically, a moderate correlation was detected in the sensory factors that analyze the processing of the vestibular system (movement for the SP-2 and balance & motion for the SPM), the proprioceptive system (body position for the SP-2 and body awareness for the SPM) and oral processing (oral for the SP-2 and taste & smell for the SPM) as well as a strong correlation for the auditory system (auditory for the SP-2 and hearing for the SPM), visual system (visual for the SP-2 and vision for the SPM) and tactile system subscales (touch for the SP-2 and the SPM).

The literature indicates the need to verify that instruments used to analyze the same problem can detect the same difficulties in each individual [21]. Parham et al. [11] indicated the importance of having convergent validity studies between the SPM and other measures that are part of a comprehensive assessment to provide more evidence for the use of this sensory processing assessment. Accordingly, this research study conducted a convergent validity analysis between two sensory questionnaires commonly used clinically and in research for Spanish-speaking populations.

The results of this study are similar to those obtained in the original convergent validity studies [10] with children from the United States of America between the SPM and SP or Short-SP where the correlations between sensory factors that measure the same sensory system were moderate or strong. However, the present study obtains stronger Pearson correlations than the original study on the factors that analyze auditory, visual and vestibular systems when comparing the SPM and the SP-2 and in the auditory, visual and vestibular factor when comparing the SPM and the SP-2.

Similar studies have analyzed the convergent validity between SPM and SP, and they have obtained similar (moderate and strong) correlations between sensory factors that analyze the same sensory system [22,23]. These studies show stronger correlations in the auditory, visual, tactile [22,23] and vestibular sensory factors than our study [23]. Furthermore, Hansen and Jirikowic [24] analyzed the differences in the sensory processing of children with Fetal Alcohol Spectrum Disorders and found strong correlations between SPM and SSP subscales, showing that both measures evaluated similar constructs.

Perhaps one of the reasons that our study did not obtain strong correlations across all factors is due to the fact that although both questionnaires are available in Spanish, the SPM did not undergo cognitive comprehensibility interviews in the Spanish (Spain) population. Previous studies conducted in Spain using sensory questionnaires [25,26] show that a cultural adaptation that includes comprehensibility interviews is necessary to ensure proper understanding of the text in the target population. There may be specific questions in both questionnaires that are not correctly understood, or questions written in the negative that may lead to false negatives and ambiguities in the answers. In the test–retest reliability analysis of the Adolescent/Adult SP [27] sensory questionnaire for children aged 11 and older in Spain, Gándara-Gafo et al. [26] showed that questions written in the negative showed worse repeatability scores.

Although the SP-2 does not have any scale to analyze social participation and planning & ideas, these scales have obtained a moderate or strong correlation with the conduct, social–emotional and attentional abilities subscales of the SP-2, highlighting the strong connection between attention and planning & ideas. These results are similar to those obtained in the original construct validity studies between SPM and SP [11] in which the authors [11] report that these results are expected since praxis and social participation reflect higher level integrative abilities that have a direct impact on adaptative behavioral functioning.

The analysis of difficulties in sensory processing and integration and the impact on occupational participation is increasingly considered in the assessment and intervention of children with a wide range of developmental vulnerabilities. In Spain, more and more occupational therapists are working with children with sensory integration problems, making it necessary to have tools with adequate psychometric data. The main limitation of this study is the convenience sample used from a private pediatric occupational therapy center in Spain. However, this study provides occupational therapists with information on these two widely used sensory questionnaires. The results of this study support clinical use of either the SP-2 or the SPM with the Spanish population given that both tools correlate adequately in those subscales that measure the same sensory factor.

It should be noted that a new version of the SPM [12] has recently been published; however, the present study was already in progress at the time of its publication. Future research should analyze the convergent validity between the SP-2 and the SPM-2. 

## 5. Conclusions

This study concludes that there is a moderate to strong convergent validity between the SP-2 and SPM for children with sensory integration issues aged between 5 and 12 years. Our results indicate that Spanish clinicians can use either sensory questionnaire when assessing sensory integration problems in this population.

## Figures and Tables

**Table 1 children-10-01516-t001:** Characteristics of the sample (n = 116). Results expressed in number (percentage of the total sample).

Variable		Sample (n = 116)
Gender	Female	36 (31)
	Male	80 (69)
Diagnosis	None	79 (68)
	ASD	16 (13.8)
	ADHD	11 (9.5)
	High Capacities	2 (1.8)
	Others	8 (6.9)
Age group	5 years	18 (15.5)
	6 years	26 (22.4)
	7 years	24 (20.7)
	8 years	22 (19)
	9 years	8 (7.8)
	10 years	7 (6)
	11 years	7 (6)
	12 years	3 (2.6)

Note: ASD = Autism Spectrum Disorder; ADHD = Attention Deficit Hyperactivity Disorder.

**Table 2 children-10-01516-t002:** Convergent validity of the Sensory Profile-2 with the Sensory Processing Measure (n = 116).

SPM/ SP-2	Social Participation	Vision	Hearing	Touch	Taste & Smell	Body Awareness	Balance & Motion	Planning & Ideas
	R (p)	R (p)	R (p)	R (p)	R (p)	R (p)	R (p)	R (p)
**Auditory**	0.30 (0.001)	0.50 (<0.01)	0.67 (<0.001)	0.50 (<0.001)	0.37 (<0.001)	0.36 (<0.001)	0.37 (<0.001)	0.37 (<0.001)
**Visual**	0.23 (0.013)	0.63 (<0.001)	0.54 (<0.001)	0.39 (<0.001)	0.24 (0.008)	0.35 (<0.001)	0.36 (<0.001)	0.34 (<0.001)
**Movement**	0.37 (<0.001)	0.57 (<0.001)	0.44 (<0.001)	0.41 (<0.001)	0.13 (0.174)	0.53 (<0.001)	0.45 (<0.001)	0.50 (<0.001)
**Touch**	0.34 (<0.001)	0.53 (<0.001)	0.52 (<0.001)	0.67 (<0.001)	0.39 (<0.001)	0.45 (<0.001)	0.38 (<0.001)	0.40 (<0.001)
**Body Position**	0.34 (<0.001)	0.49 (<0.001)	0.39 (<0.001)	0.42 (<0.001)	0.23 (0.013)	0.40 (<0.001)	0.56 (<0.001)	0.46 (<0.001)
**Oral**	0.22 (0.019)	0.22 (0.016)	0.22 (0.018)	0.30 (0.001)	0.43 (<0.001)	0.20 (0.035)	0.28 (0.002)	0.15 (0.107)
**Conduct**	0.45 (<0.001)	0.46 (<0.001)	0.41 (<0.001)	0.39 (<0.001)	0.30 (0.001)	0.55 (<0.001)	0.42 (<0.001)	0.39 (<0.001)
**Social Emotional**	0.30 (0.001)	0.37 (<0.001)	0.36 (<0.001)	0.28 (0.002)	0.19 (0.045)	0.29 (0.002)	0.35 (<0.001)	0.30 (0.001)
**Attentional**	0.44 (<0.001)	0.56 (<0.001)	0.45 (<0.001)	0.44 (<0.001)	0.15 (0.103)	0.48 (<0.001)	0.44 (<0.001)	0.62 (<0.001)

Note. SPM = Sensory Processing Measure. SP-2 = Sensory Profile-2.

## Data Availability

The data are available on request from the corresponding author. The data are not publicly available due to privacy aspects.

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
