# Peer review of "Convergent Validity of Two Sensory Questionnaires in Spain: Sensory Profile-2 and Sensory Processing Measure"

_children, 2023, doi:10.3390/children10091516_

Round 1

Reviewer 1 Report

This manuscript offers some interesting information for clinicians in Spain using a parent report measure of sensory processing. There are minor revisions that would make the manuscript more readable and accurate. First and foremost is to make clear to the reader that both of the measures presented in this study are 'screening' tools and should not be considered 'assessments'. Rather they should be used as part of a comprehensive assessment that includes an examiner administered scale along with parent report questionnaires. 

Introduction line 63, authors refer to Bundy and Lane , however the SPM predates the publication of their book. Please revise the statement or change the reference. 

Line 84 use of the word 'indistinctly' is unclear. Please revise

Participants

Line 109 refers to inclusion as having parent consent but earlier authors say consent was waived. Please be consistent 

Measures

Line 112 the word you want to use is analyzes and analyses

Same is true in line 124

Line 122-123 is a repetition of a same sentence as lines 116-117

Procedures

This section needs to be completely re-organized. The first two paragraphs are not specific to Procedures and rather they relate to the justification of the methodology and the way the data were interpreted

Results

Please explain the number in the parentheses in Table 1

Table 2 needs to be reformatted based on APA criteria

Discussion

Line 185 ...remove issues and use differences or variation instead

Line 190 remove assessment and replace with screening tool or part of comprehensive assessment

Lines 193 - 197 entire paragraph appears redundant , either revise or remove

Lines 237-238 please revise the language ...words are confusing, "superior functions", "adaptive conductive functioning"

Last 3 paragraphs the word interchangeably is used over 3 times. please try to find another way to convey the benefits of the two measures. 

I recommend that the entire manuscript be reviewed for use of the english language and english grammar.

Author Response

This manuscript offers some interesting information for clinicians in Spain using a parent report measure of sensory processing. There are minor revisions that would make the manuscript more readable and accurate. First and foremost is to make clear to the reader that both of the measures presented in this study are 'screening' tools and should not be considered 'assessments'. Rather they should be used as part of a comprehensive assessment that includes an examiner administered scale along with parent report questionnaires.

Thank you for your comment. We have clarified the text for better understanding (Line 48-50 and Line 214-215). 

Introduction line 63, authors refer to Bundy and Lane , however the SPM predates the publication of their book. Please revise the statement or change the reference. 

Thank you for your comment. Reference has been changed (Line 75)

Line 84 use of the word 'indistinctly' is unclear. Please revise

Done

Participants

Line 109 refers to inclusion as having parent consent but earlier authors say consent was waived. Please be consistent 

Thank you for your comment. The Material and Methods section has been modified to clarify this Section.

Measures

Line 112 the word you want to use is analyzes and analyses

Done

Same is true in line 124

Done

Line 122-123 is a repetition of a same sentence as lines 116-117

Thank you for your observation. One of them have been removed.

Procedures

This section needs to be completely re-organized. The first two paragraphs are not specific to Procedures and rather they relate to the justification of the methodology and the way the data were interpreted

Thank you for your comment. We have modified the text for better understanding to the indicated sections (Line 48-50; Line 88-94; Line 110-112; Line 152-153). 

Results

Please explain the number in the parentheses in Table 1

Done

Table 2 needs to be reformatted based on APA criterio

Done

Discussion

Line 185 ...remove issues and use differences or variation instead

Done

Line 190 remove assessment and replace with screening tool or part of comprehensive assessment

Done

Lines 193 - 197 entire paragraph appears redundant , either revise or remove

Done (remove)

Lines 237-238 please revise the language ...words are confusing, "superior functions", "adaptive conductive functioning"

Done

Last 3 paragraphs the word interchangeably is used over 3 times. please try to find another way to convey the benefits of the two measures. 

Done

Author Response

This study was to examine the concurrent validity between sensory profile-2 and sensory processing measure in children with sensory integration problems. You have completed an important study. Excellent! However, it is judged that some modifications are needed in its description. It is as follows.

Convergent validity and concurrent validity are distinctly different in terms of their intended outcomes. However, there appears to be a discrepancy between the usage of "convergent validity" in the title and "concurrent validity" in the research purpose within the abstract. This study requires clarification regarding its intended focus.

Done

The abstract is concise. While it's acceptable to keep the research title brief and omit the research subject, it is essential to specify the subject within the abstract. In your abstract, you mentioned that the study involved Spanish children, but it's important to outline the specific characteristics of these children. Additionally, the research subject should be reiterated in the conclusion.

Done

Introduction: In the introduction, ensure alignment between the research purpose, research title, and introduction. Provide more comprehensive details about sensory integration problems in the introduction.

Thank you for your comment. We have add more information about sensory integration in the introduction (Line 38-41), the Validity (Line 88-94) and the research subject (Line 99-103)

Method: Elaborate on the criteria used for selecting and excluding research subjects. Incorporate the clinical characteristics of the study subjects.

Thank you for your comment. We have add more information about exclusion criteria (Line 132) and the clinical characteristics of the sample in Table 1.

Results: Conduct an analysis of the total score correlation between the two evaluation tools.

Thank you for your comment. The correlation of the total score between both tools is not measurable. The SP-2 does not have a total score, which makes it impossible to compare total score of both tools.

Discussion: Commence your review by summarizing the key findings of this study.

Done

Upon examining Table 2, it becomes evident that the study's results do not demonstrate moderate to strong correlations between the two assessment tools. Please elucidate the basis upon which this judgment is made

Thank you for your comment. Lalinde et al., (2018) indicates a moderate correlation with 0,30-0,49 scores and strong correlation with scores above at 0,5. Following the methodology indicated by Lalinde et al., (2018) the correlation found in this study are moderate to strong in the factors that analized the same Sensory sistem. We have add more information (Line 162-169 and 203-205).

Round 2

Reviewer 2 Report

You put in diligent effort your thesis.

Nevertheless, there is an unfortunate issue present in the revised manuscript.

Given that the participants of this study were children with sensory problems, the generalization of this study can only be extended to individuals within the population that adhere to the defined selection and exclusion criteria. This aspect should be appropriately acknowledged in the conclusion of the study. 

I anticipate that this matter will find application in broader academic research endeavors. 

Author Response

You put in diligent effort your thesis.

Nevertheless, there is an unfortunate issue present in the revised manuscript.

Given that the participants of this study were children with sensory problems, the generalization of this study can only be extended to individuals within the population that adhere to the defined selection and exclusion criteria. This aspect should be appropriately acknowledged in the conclusion of the study. 

I anticipate that this matter will find application in broader academic research endeavors. 

Thank you for your comment. We have done some modifications in the conclusions.

"This study concludes that there is a moderate to strong convergent validity between the SP-2 and SPM for children with sensory integration issues aged between 5 and 12 years. Our results indicate that Spanish clinicians can use either sensory questionnaires when assessing sensory integration problems in this population".